# Urban environmental and population factors as determinants of COVID-19 severity: A spatially-resolved probabilistic modeling approach

Jacob Roxon[1], Marie-Sophie Dumont[1,2], Eric Vilain[1,3], Mircea T. Sofonea[4], Roland J-M. Pellenq[1,5]*

**1** EpiDaPo Lab - CNRS/George Washington University Children's National Medical Center, Children's Research Institute, Washington, District of Columbia, United States of America, **2** iGLOBES laboratory, CNRS, ENS-Paris/PSL Université and the University of Arizona – Marshall Building, Tucson, Arizona, United States of America, **3** School of Medicine, UC-Irvine, Irvine, California, United States of America, **4** Pathogenesis and Control of Chronic and Emerging Infections (PCCEI, U1058) – University of Montpellier, INSERM, EFS, Univ. Antilles and CHU de Nimes, Montpellier, France, **5** Institut Européen des Membranes, CNRS and University of Montpellier, Montpellier, France

* roland.pellenq@cnrs.fr

## Abstract

COVID-19 is caused by a severe acute respiratory syndrome due to the SARS-CoV-2 coronavirus. It has reshaped the world with the way our communities interact, people work, commute, and spend their leisure time. While different mitigation solutions for controlling COVID-19 virus transmission have already been established, global models that would explain and predict the impact of urban environments on the case fatality ratio CFR of COVID-19 (defined as the number of deaths divided by the number of cases over a time window) are missing. Here, with readily available data from public sources, we study the CFR of the coronavirus for 118 locations (city zip-codes, city boroughs, and cities) worldwide to identify the links between the CFR and outdoor, indoor and personal urban factors. We show that a probabilistic model, optimized on the sample of 20 districts from 4 major US cities, provides an accurate predictive tool for the CFR of COVID-19 regardless of the geographical location. Furthermore, we show that the validity of the model extends to other infectious diseases such as flu and pneumonia with pre-COVID-19 pandemic data for 3 US cities indicating that the first COVID-19 wave severity corresponds to that of pneumonia while other COVID-19 waves have the severity of influenza.When adjusted for the population, our model can be used to evaluate risk and severity of the disease within different parts of the city for different waves of the pandemic. Our results suggest that although disease screening and vaccination policies to containment and lockdowns remain critical in controlling the spread of airborne diseases, urban factors such as population density, humidity, or order of buildings, should all be taken into consideration when identifying resources and planning targeted responses to mitigate the impact and severity of the viruses transmitted through air.

**Data availability statement:** All data are in the manuscript and/or Supporting Information files.

**Funding:** This work was supported by CNRS-Innovation with a pre-mat grant (2021-2022 to JR). The funders had no role in study design, data collection and analysis, decision to publish, or preparation of the manuscript.

**Competing interests:** The authors have declared that no competing interests exist.

## Authors summary

The Sars-Cov2 virus has caused significant disease in humans inducing the COVID19 crisis. Sars-Cov2 virus is airborne transmitted among humans through mucus laden expectorated droplets. Currently, the amount of virus per droplet is unknown but the role of relative humidity is documented as well as that regarding urban population concentration. Here, we propose a multiscale statistical model that can predict the COVID19 case fatality ratio that characterizes the disease intensity in urban environment from the zip-code level up to the global city scale. We found a simple equation based on 5 urban descriptors and a single constant factor called $CFR_0$ that is the disease CFR corrected for the urban environment and urban population profile. Hence $CFR_0$ is location independent. The statistical weight for each urban and population descriptor is adjusted once on COVID19 CFR borough data for 4 major US cities and then used for all other scales and other cities around the globe with no further adjustment with a very good level of accuracy showing large variations within a given city and from city to city. The urban descriptors were ranked in 3 categories: outdoors, indoors and personal. The dominating "outdoors" one is the relative humidity with a smaller influence of the urban form while "indoors" descriptors (number of housing units per building and number of dwellers per unit) are nearly equal to 50%; the only important "personal" parameter is the population older than 65 and not income level. We found that the COVID29 $CFR_0$ is 10 times larger than that for the flu and comparable to that of pneumonia. Our study provides new insights into the deployment and intensity of the COVID19 disease in different urban environments.

## Introduction

Towards the end of January 2020, the World Health Organization (WHO) declared a Public Health Emergency of International Concern, which within less than 6 weeks, on March 11th 2020 was described as a global pandemic of coronavirus (COVID-19) disease. The most severe impact of the COVID-19 has been observed in urban areas [1,2], which in most countries are home to over 70% of population. And while the implementation of pandemic response measures, such as vaccinations or social distancing and face covering restrictions, have provided a path towards controlling the epidemic [3,4], they may not be sufficient to end the disease, which is likely to become endemic [5] resulting in infections and mortalities, severity of which, seen through the prism of the Case Fatality Ratio, CFR, (i.e., the ratio of the number of deaths to the number of confirmed cases) changes from country to country (Fig 1).

In the context of COVID-19, crisis management has involved dynamic assessments of evolving situations, adaptation to fast-changing circumstances, and coordinated actions among diverse stakeholders. Within this framework, patterns of collaborative actions have emerged, which guided by interactionist theories, aimed at achieving expected outcomes while navigating through unstructured collaborative processes. A dynamic interplay of individuals, groups, and organizations shape the

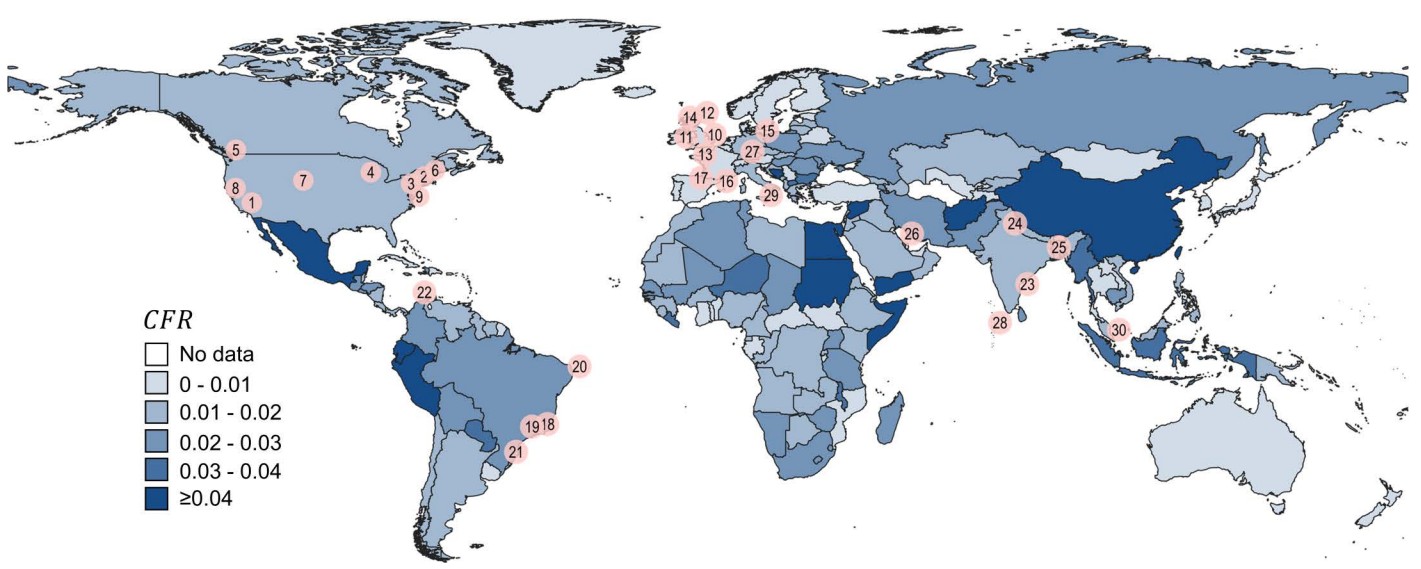

**Fig 1. The severity of the COVID-19 throughout the world showing cumulative COVID-19 CFR (S1 Table) at the country scale between 01/2020 and 01/2022 with a geo-location of 30 cities (S2 Table) used in this study. Map created using the Free and Open Source QGIS.** Copyright-free countries boundaries data were taken from the world bank (https://datacatalog.worldbank.org/search/dataset/0038272).

collective behavior found at the core of interactionist theories. These theories highlight the importance of social interactions, shared understandings, and emergent behaviors in driving collaborative societal responses. Rather than relying solely on hierarchical structures, interactionist perspectives recognize the need for flexible and adaptive approaches that allow for decentralized decision-making and emergent solutions. Patterns of collaborative actions in managing large scope crisis, such as COVID-19, involve multi-level coordination among government agencies, healthcare organizations, businesses, community groups, and individuals. These collaborative efforts encompass a range of activities, including information sharing, resource allocation, policy development, public communication, and implementation of preventive measures towards protecting vulnerable population. However, in the midst of the COVID-19 crisis, unstructured collaborative processes, characterized by ambiguity, conflicting priorities, and decentralized decision-making left our society exposed and vulnerable to the virus spread. Interactionist theories offer insights into how individuals and groups negotiate meaning, establish social norms, and adapt to changing circumstances. Flexible, adaptive approaches that emphasize communication, trust-building, and shared goals are essential for effective collaboration amidst uncertainty. For example, Awareness Information Network (AIN) approach serves as a platform for facilitating collaboration and information exchange among stakeholders involved in crisis management. AIN enables real-time communication, data sharing, and coordination of response efforts across organizational boundaries. Semiotic validation plays a crucial role in ensuring the clarity, consistency, and effectiveness of communication within the AIN, minimizing misinterpretation and fostering trust among stakeholders. In sum, effective crisis management in the context of a large intensity crisis such as the COVID-19, requires a dynamic, collaborative approach informed by interactionist theories and utilizing tools such as Awareness Information Network offers a means of navigating through unstructured collaborative processes [6,7]. And while recent literature on spatially resolved COVID-19 epidemiology mostly focuses on simulating the spread the disease in time at the scale on countries such as in [8,9] or at the scale of counties such as in [10] using Suspected/Infected/Recovered (SIR) differential coupled equations approaches, to our knowledge, this work offers a unique perspective to provide quantitative insights on spatially resolved epidemiology at the localized urban scale of zip codes, while including physical parameters and urban descriptors.

SARS-CoV-2 virus stability is affected by physical, chemical, and biological factors [11]. On inanimate surfaces, the most important are surface type, relative humidity [12], temperature [13], moisture content of the suspending medium (mucus). Although among humans, there are different possible transmission routes of COVID-19, the focal point of scientists has been airborne propagation of the SARS-CoV-2 [14], which involves projection of virions in aerosolized droplets of mucus from an infected individual, inhaled by sane individual. Due to air drag and gravity, virus-laden droplets with diameter $d \geq 5\mu m$ emitted in a cough or sneeze, have a generally low projection distance of less than few meters [15,16] with a short persistence duration. Smaller virus-laden particles ($d < 5\mu m$), can remain in air for hours and consequently be dispersed through airborne route both indoors and outdoors. In addition, the propagation of COVID-19 has been linked to particulate matter (PM) pollution levels showing a positive correlation between long-term exposure to high concentration of PM particles and the number of COVID-19 deaths [17]. At a nanoscale, the viral load and virulence through the SARS-CoV-2 spike proteins and ACE2 human cells receptors fusion mechanisms, correlate directly with infectivity, disease phenotype, morbidity, and mortality, which should be accounted for correlating conventional aerosol metrics, such as PM2.5 or PM10 concentrations to COVID-19 cases, or deaths, and provide a direct explanation of the variability of the COVID-19 severity through the Case Fatality Ratio [18]. CFR has also been linked to the viremia [19,20], a measure of the quantity of virions emitted by infected patients in the form of aerosolized droplets that can circulate both indoors and outdoors. Severe forms of COVID-19 lead to respiratory failure, shock or multiple organ failures, or even death. Although, CFR is considered a crucial quantity to predict emergency and health care infrastructure deployment to manage the spread of COVID-19, understanding of mechanisms that in an urban setting contribute to the COVID-19 severity and the airborne propagation of SARS-CoV-2 remains limited, subsequently preventing accurate determination of COVID-19 CFR at the scale of neighborhoods, cities or even countries [21–30]. Here, we propose a probabilistic model to statistically predict CFR based on 3 distinctive Urban Factors ($U_i$): Personal ($P_i$), Indoor ($I_i$), and Outdoor ($O_i$). We find that while the approach works best for city-wide predictions ($N = 30$ cities worldwide), we can also successfully utilize it at other scales, ranging from zip codes or city districts (S2 Fig) to county or even country levels.

## Results

The main component of the CFR model relies on the data from the first year of the pandemic for 30 cities worldwide (Fig 1, S1 Table) at various geospatial scales. Since at the very beginning of the pandemic there were major challenges in reporting values of infected people, $C19_c$, we chose a minimal time window of six months, which is sufficient to study the first wave of the COVID-19 virus and investigate CFR correlations between $C19_c$ and the number of deaths, $C19_d$ in the form:

$$C19_d = a \times C19_c^k \tag{1}$$

where, $a$ is an algebraic proportionality constant and $k$ is an exponent factor, which for COVID-19 has been $\cong 1$ (S4 Fig). We began evaluating $a = CFR = C\frac{19_d}{C}19_c$ at a local scale for city wards and districts (S2 Fig, S2–S3 Tables) in 4 US cities: New-York NY (Fig 2a–2b), Los Angeles CA, Chicago IL and Seattle WA, during the first COVID-19 wave ($\lambda_1$ factor) from March 2020. Although, similar data for the wards of Washington D.C. is available and has been used throughout this study, we chose not to use it in the model optimization process due to low (<100,000) population of D.C. wards but rather keep this information for validation purpose at the wards scale. While for most regions available data (S2 Table) extends beyond the first 6–12 months of the pandemic and captures multiple waves, herein defined $\lambda_1$ captures the earliest and most severe time frame of the pandemic. Upon investigating time series of CFRs, we find that during $\lambda_1$ they resemble sigmoid functions. To eliminate time component, we use average $CFR(\lambda_1)$ obtained directly from upper plateau range of CFR distribution (Fig 2a). Although, one can use the steepest slope (Fig 2b) or the maximum value (Fig 2a) of CFR distribution to quantify its severity cities districts and cities ($N = 50$, Fig 2c–2d), $CFR(\lambda_1)$ provides the most stable averaging approach that works with inconsistencies and fluctuations in raw COVID-19 data. With such defined $CFR(\lambda_1)$, we proceeded to

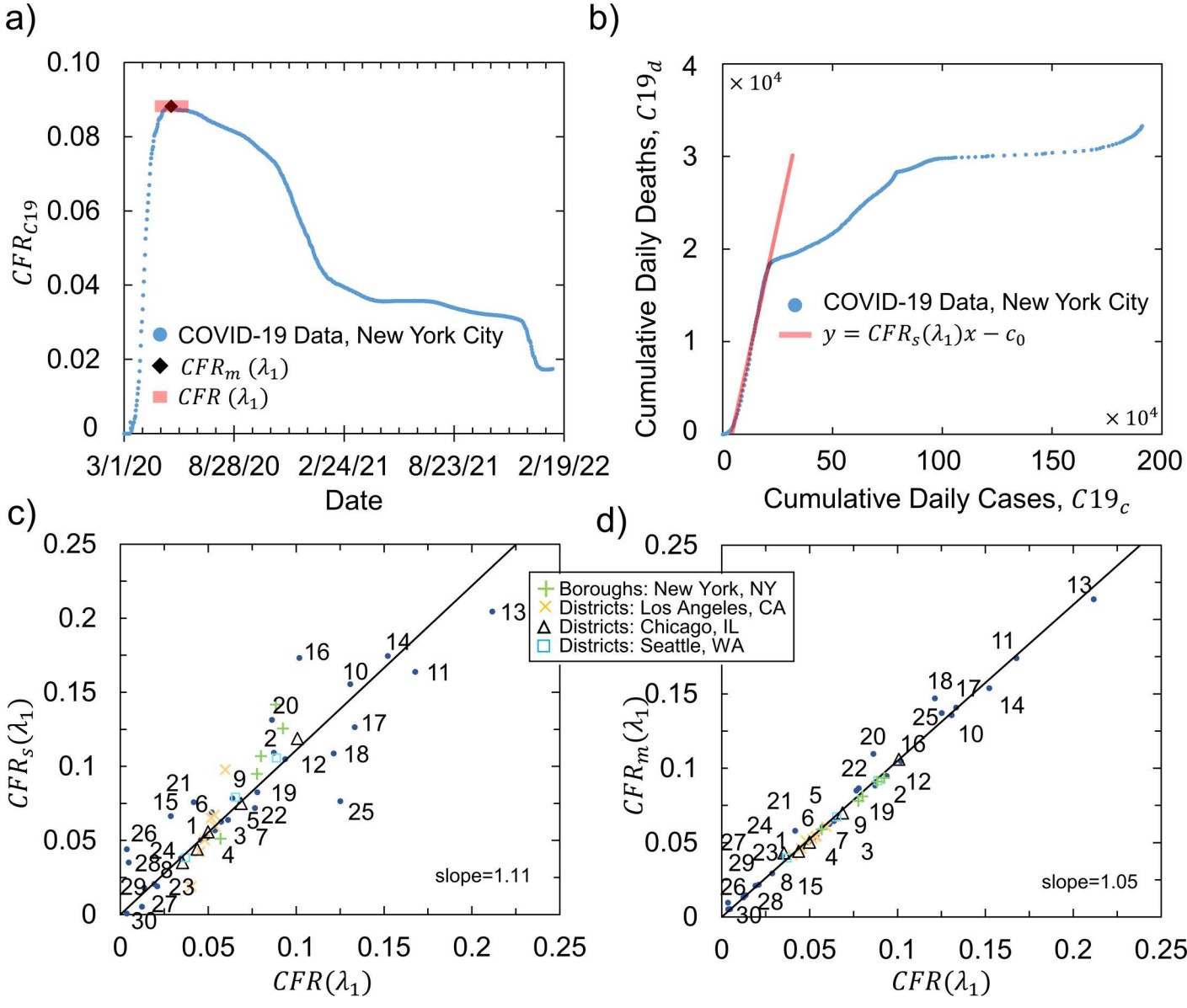

**Fig 2. COVID-19 CFR methodology showing (a) CFR time series with examples for calculating** $CFR(\lambda_1)$ **and** $CFR_m(\lambda_1)$ **and (b)** $CFR_s(\lambda_1)$ **using Eq. 1 for the first wave of the pandemic in New York City, NY.** Linear correlation between (c) $CFR(\lambda_1)$ vs. $CFR_s(\lambda_1)$ and (d) $CFR(\lambda_1)$ vs. $CFR_m(\lambda_1)$ for city districts ($N = 20$, S4 Table) and cities ($N = 30$, S5 Table).

define Personal ($P_i$), Outdoor ($O_i$) and Indoor ($I_i$) Urban Factors, $U_i$s (Fig 3a), where $i$ represents a geographical location, which to formulate the model is limited to wards or districts of cities.

The personal urban factor ($P_i$) is defined by proportion of the population older than 65-year-old, which encompasses wealth and health data, while also aggregating comorbidities that prevail in in this age group (see SI). The indoor urban factor ($I_i$, see SI) captures the housing conditions with metrics explained by the likelihood of human interactions due to living conditions. Finally, the outdoor urban factor ($O_i$, see SI) describes the influence of city texture and weather conditions through the pressure-temperature dependent thermodynamic quantity – relative humidity.

PLOS Digital Health

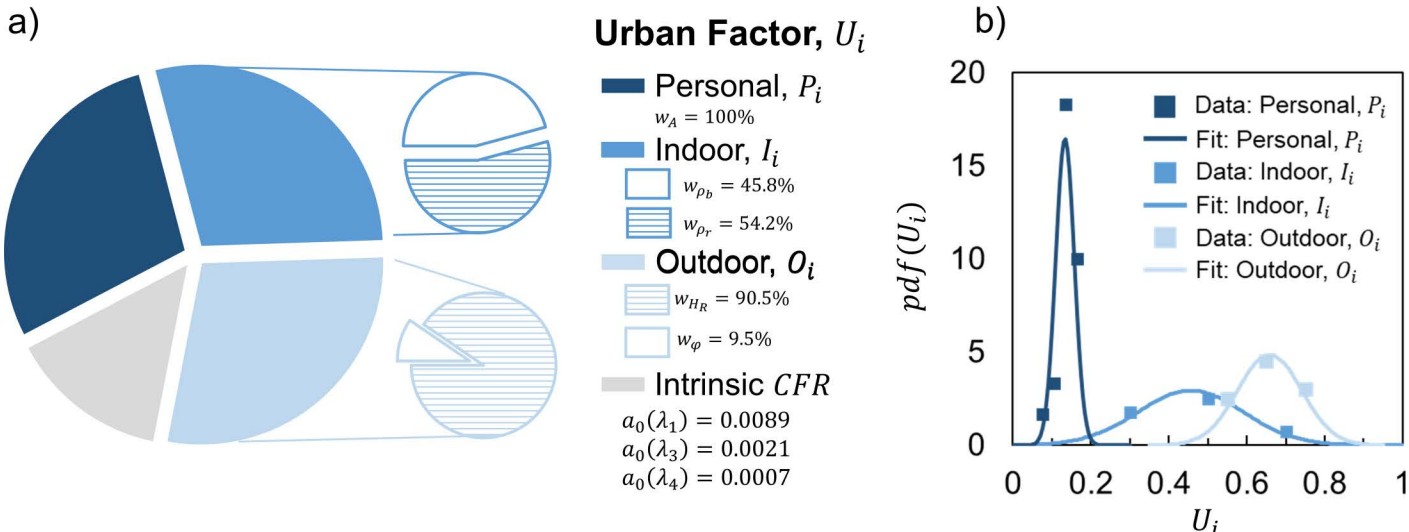

**Fig 3. (a) Urban Factors with intrinsic CFR values for different waves of the pandemic and model weight parameters optimized for the city districts (SI, S4 Table) with (b) their probability distributions.** The drop in $a_0$ values through the different COVID-19 waves is the clear consequence of vaccination combined to social distancing measures. Interestingly enough, the of the fourth wave ($\lambda_4$) is that of the regular influenza disease (see S8 Table).

To identify differences in the COVID-19 spread in each location as captured with CFR, we adopted city texture methodology (see Materials and Methods, SI), among other variable used in the determination of Urban Factors (S4–S8 Tables). However, for the calibration of weight parameters in Eqs. 3–5 only city district ($N = 20$) from S3 Table have been used to allow a sufficient sample of uncorrelated data from various geographical regions to be used for validation of the model ($N = 118$ for cities, zip codes and boroughs, including 9 outliers).

To correct measured CFR values, the dimensionless weight factors derived from $CFR(\lambda_1)$ for 20 city districts from 4 major US cities were adjusted using a theoretical model $a_i$, defined as the $CFR(\lambda_1)$ for region $i$ during time window $\lambda_1$ (first pandemic: March-August 2020, Fig 4a, see SI for the derivation of Eq. 2):

$$a_i = \frac{a_0}{[1 - P_i][1 - I_i][1 - O_i]}$$

(2)

where, $a_0$ is a constant indicative of intrinsic $CFR(\lambda_1)$ (corrected for the urban factors under the form of a product of 3 independent probabilities) during the initial, first wave of the pandemic with no, or minimal, city-wide social distancing and vaccine protocols. Thus, $a_0$ becomes $CFR(\lambda_1)$ that corresponds to natural human response to COVID-19 virus corrected for all type of Urban Factors, a correction derived using independent and uncorrelated $U_i$s (Fig 3b): Personal, Indoor and Outdoor Factors, where each $[1 - U_i]$ component reflects the correction of measured $CFR(\lambda_1)$. From the distribution of factors, it can be concluded that age index, captured by $P_i$, has much lower impact on $CFR(\lambda_1)$ than $O_i$ or $I_i$, where their mean values closer to 1 have greater influence over $CFR(\lambda_1)$. With a very low absolute residual sum between predicted and measured CFR, which is 15% of $\sum_{i=1}^{n=20} CFR_i(\lambda_1)$ (SI) built upon 20 city districts, the $CFR(\lambda_1)$ model results in $a_0 = 0.0089$, which at the global scale is lower than previously measured intrinsic $a_0(\lambda_1) = 0.014$ (22) for single geographical region. While real-time predictability of $a_0$ can be improved by considering a day-to-day and local variability of relative humidity, here it is assumed to be an external constraint affecting the whole urban area using annual average values. To adequately capture risk and severity of $CFR(\lambda_1)$ at the local scale of cities, for zip-code, wards, districts and boroughs

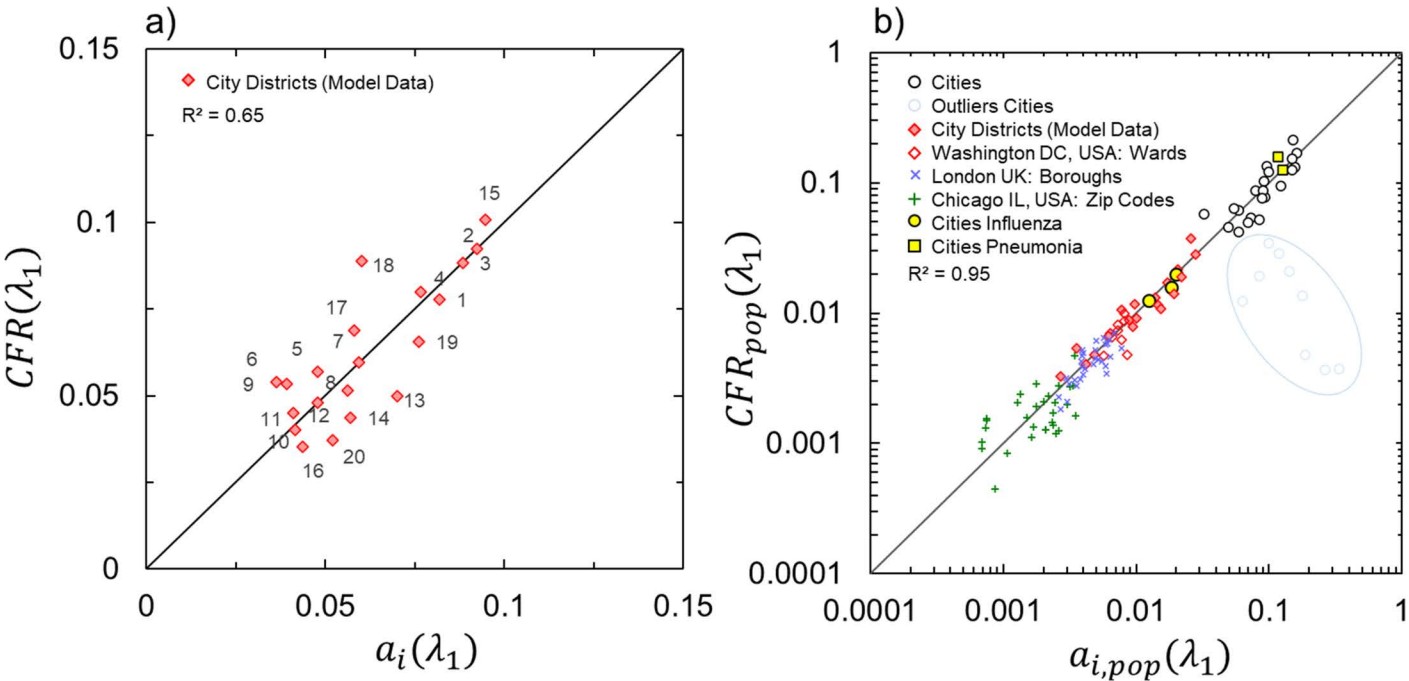

**Fig 4. Urban Factors model predictions showing (a) comparison between measured and predicted $CFR(\lambda_1)$ from Eq. 2 for 20 city districts for 4 major US cities (S2 Fig, S4 Table) between 03/2020-08/2020 used for optimizing urban factor weight parameters and intrinsic CFR.** Linear fitting with slope coefficient of unity provides $R^2 = 0.65$ and $RMSE = 0.012$. (b) Comparison between measured and predicted population adjusted $CFR_{pop}(\lambda_i)$ for 118 location worldwide (S4–S8 Tables) during different waves of the pandemic (excluded 9 outliers for cities in S5 Table for the correlation calculations). Linear fitting with slope coefficient of unity provides $R^2 = 0.95$ and $RMSE = 0.31$ in the log-log scale. Predicted CFR and $CFR_{pop}$ values used the same urban factor weight parameters in both figures. $CFR_{pop}$ for influenza (3 US cities, see S8 Table in SI) and community-acquired pneumonia (2 US cities, see S8 Table in SI) from 2018 is showed in comparison to COVID-19 $CFR_{pop}$.

values we multiple measured and predicted case fatality ratio values by the fraction of the population of a specific city (Fig 4b). For cities taken as a whole, this ratio remains unity.

## Discussion

The ability of our model to extend beyond city districts scale was then demonstrated at the global city scale with no adjustment of urban factor weight and $a_0$ parameters for 30 cities and 88 smaller scale locations (i.e., zip-codes, wards) worldwide. Regardless of the scale we find that very accurate population adjusted $CFR_{pop}(\lambda_1)$ predictions ($R^2 = 0.95$) can be made for any place in the world, provided that no extreme countermeasures were adopted at the beginning of the pandemic. Furthermore, the Urban Factors model from Eq. 2 can be applied to make prediction over different time scales of the pandemic. With additional cumulative $C19_C$ and $C19_d$ data between the start of the pandemic and third wave (March 2020 - June 2021) and forth wave (March 2020 - January 2022) respectively, and keeping all $w's$ parameters constant (Fig 2), we could solely adjust $a_0(\lambda_i)$ to $a_0(\lambda_3) = 0.002$ and $a_0(\lambda_4) = 0.001$ respectively while maintaining acceptable accuracy of predictions. The sharp decline in $a_0(\lambda_i)$ values over time reflects the immense effect of the vaccine campaign on the severity of the pandemic and the transition to virus variants being less harmful in terms of severe cases (i.e., deaths). The variability in $a_0(\lambda_i)$ is expected as over the course of the entire pandemic, with yearly average $H_R$ values, city layouts and census data exposed to minimal changes, $w$ weight parameters should not vary to a noticeable extent. It is also important to note that the model significantly overestimated $CFR(\lambda_1)$ for 9 cities (Fig 4a, S5 Table) where measured $CFR(\lambda_1)$ were very close to $a_0(\lambda_1)$. Upon further investigation, these cities were found to be either isolated territories

(such as Maldives or Malta, which, based on size and population, are treated as cities) or quick to adopt social distancing and/or lockdown measures (as in San Francisco CA, Berlin in Germany, or Singapore) during the first wave, $\lambda_1$, of the COVID-19 pandemic. We can conclude therefore that measured CFR as predicted by the coronavirus severity equation (Eq. 2) reflects the absence or inefficiency of sanitary measures taken by local and/or national authorities to manage the first wave of COVID-19.

Multi-spatial scale application of the model with no adjustment to its weight factors allows us to study the impact of specific variables that make up the three distinctive urban factors. Relative humidity takes 90.5% of $O_i$, while $\varphi$ (characterizing city texture, see SI) is responsible for the remaining 9.5%. High percentage value of $H_R$ supports the airborne nature of pandemic transmission underlying the role of evaporation and thermodynamic conditions of the surrounding atmosphere [11,13]. The effect of city texture parameter $\varphi$ can be linked to persistence of active outdoor forms of SARS-CoV-2 previously related to urbanization and particulate atmospheric pollution [17,27]. This also suggests that ordered districts in a given city (i.e., Brooklyn in NYC) are more likely to retain laden droplets than less ordered ones (i.e., Bronx in NYC). Indoor Factor, defined by the household size and density of units have similar contribution, 54% and 46%, respectively, which can be explained by the close-range contaminations and circulation of the virus not just inside units, but also in between them due air fluxes and recirculation [25,26] and its presence on common surfaces and spaces such as elevators, stair cases, lobbies [8,27]. Since the higher density of inhabitants per housing unit results in higher probability of CFR, this work reveals an aspect of social discriminations linked to wealth and living standards. Finally, the Personal Factor is derived solely from population older than 65 years old, which suggests that the current healthcare systems in place across the 30 cities we considered, were able to respond to patient's needs regardless of their income.

Reduction in global $a_0$ by a factor 4 (from 0.0089 to 0.0022) during the first three waves followed by a further reduction by 2.8 between the third and fourth waves (from 0.0022 to 0.0008) reflects the joint positive impact of improved symptomatic care, non-pharmaceutical interventions (though preventing critical care overload), early detection, vaccination, monoclonal antibodies, anti-viral treatments and eventually the spread of the Omicron variant. Note that the value of $a_0$ for the fourth wave of COVID-19 is close to that of the influenza disease (see S8 Table in SI) Each $U_i$ changes predicted CFR according to expected norms that is an increase in relative humidity, density of urban dwellers or the proportion of population above 65 years of age, all increase CFR values. It is shown that the approach remains operative at the global city scale making model fit predictions with $R^2 = 0.95$ and a low $RMSE = 0.31$ population adjusted $CFR_{pop}$. It is interesting to note that averaging measured CFRs, across all US and UK cities considered in this work, gives comparable to cumulative at country scale values for $CFR\,(\lambda_3)$: 2.0% vs 1.8% (US) and 2.7% vs. 2.7% (UK) and CFR): 1.1% vs 1.2% (US) and 0.9% vs. 1.0% (UK) reflecting the fact that the COVID-19 pandemic is mainly an urban phenomenon.

In the field of epidemiology, it is often considered that the CFR has limited time validity (usually a few months) and is spatially restricted to a country level. This work demonstrates, however, that the CFR carries a multiscale length and time scale stability and thus not only can be used to infer disease reported severity in both real time and during the entire pandemic. It can also be averaged by location from the zip code to city scales worldwide. Therefore, predicting the apparent severity of COVID-19 using CFR provides new dynamic means for evaluating cities in real-time with changing urban factors to equip decision makers and public authorities with quantifiable data to optimize and target their seasonal response to airborne infectious diseases, such as COVID-19 at any geo-spatial scale. As a final consideration, it is interesting ot note that without adjustment of weight parameters, Eq. 2 is able to describe the influenza and pneumonia data for a few cities in the US with publicly available data (see SI). As a validation of our approach, the $a_0$ of community-acquired pneumonia is very close to that of COVID-19 first wave while that of successive COVID-19 waves is converging to the $a_0$ of influenza (i.e., 10 times less that that of pneumonia of COVID-19($\lambda_1$).

## Materials and methods

For all input data, we resort to using publicly available building footprints, weather, and census data (S1–S2 Tables). We define the first $U_i$, Personal Factor as:

$$P_i = w_A \frac{N_{A,i}}{N_{0,i}} + w_I \frac{N_{I,i}}{N_{0,i}} \{0 \le w_A, w_I \le 1 w_A + w_I = 1$$

(3)

$$P_i = \frac{w_A N_{A,i} + (1 - w_A) N_{I,i}}{N_{0,i}} \{0 \le w_A \le 1\}$$

where $w_A$ is a weight factor for inhabitants of age, $(A, i) \ge 65$ years old, $N_{A,i}$ is the total number of people $A, i \ge 65$, $w_I$ is a weight factor for an annual income, $i \le \$50,000$, $N_{I,i}$ is the total number of people with $I, i \le \$50,000$, and $N_{0,i}$ is the total number of people living in the region $i$. Note that the fraction of inhabitants older than 65-year-old strongly correlates the Healthcare Access Quality factor (HAQ, [31], see S1 Fig). The fraction of people older than 65-year-old also represents well the population dealing with comorbidities [32]. The second $U_i$, Indoor Factor is defined as:

$$I_i = w_{\rho_b} \left[ 1 - \rho_{b,i}^{-1} \right] + w_{\rho_r} \left[ 1 - \rho_{r,i}^{-1} \right] \{0 \le w_{\rho_b}, w_{\rho_r} \le 1 w_{\rho_b} + w_{\rho_r} = 1$$

(4)

$$I_i = w_\rho \left[ 1 - \rho_{b,i}^{-1} \right] + (1 - w_\rho) \left[ 1 - \rho_{r,i}^{-1} \right] \{0 \le w_\rho \le 1\}$$

where $w_{\rho_b} w_\rho$ is a weight factor for building density, $\rho_{b,i}$ (number of housing units per building), and $w_{\rho_r}$ is a weight factor for resident density $\rho_{r,i}$ (household size, which is the number of residents per housing unit) in region $i$. The final $U_i$, Outdoor Factor is calculated as:

$$O_i = w_\rho \rho_i + w_\varphi \varphi_i + w_{H_R} H_{R,i} \{0 \le w_\rho, w_\varphi, w_{H_R} \le 1 w_\rho + w_\varphi + w_{H_R} = 1$$

(5)

$$O_i = w_\rho \rho_i + w_\varphi \varphi_i + (1 - w_\rho - w_\varphi) H_{R,i} \{0 \le w_\rho, w_\varphi \le 1\}$$

where $w_\rho$ is a weight factor for the planar density of building footprints, $\rho_i$ (ratio between total area of buildings and area of region $i$), $w_\varphi$ is a weight factor for the order parameter of buildings, $\varphi_i$ [33], and $w_{H_R}$ is a weight factor for the relative humidity which is assumed constant at the city-scale, except for New York, Los Angeles and Chicago, where city district-level $H_{R,i}$ was available. The buildup surface area and the local order parameter, $\varphi_i$ reflects the local urban texture of the first shell of neighboring building with respect to a central one and is used to quantify 2-D angular order of a local urban texture obtained from the 2-D building-building pair correlation function, $g(r)$ [33], see SI), which is defined as:

$$g(r) = \frac{1}{N} \sum_{i=1}^{N} \frac{n_i(r+dr) - n_i(r)}{\rho_{city} 2\pi r \times dr}$$

(6)

where $n_i(r) \, n_i(r)$ denotes the number of buildings within the radial distance $r$ from building $i$, and $dr$ is distance increment, which for $g(r)$ calculations we chose to be 5% of the average building size, $L$. From the distribution of $g(r)$ we can directly obtain coordination number, $C_n$ (i.e., average number of neighboring buildings) using its integral in the form:

$$C_n = 2\pi \rho_{city} \int_0^{r_{min}} g(r) dr$$

(7)

The approach for selecting $C_n$ should be evaluated based on its application, which depending on desired accuracy may lead to different results. Here, the application is defining average local configuration of buildings as captured by $g(r)$ thus

leading to our use of Eq. 7, where $r_{min}$ is the first local minimum in the $g(r)$ distribution following the main peak (i.e., one with the maximum $g(r)$ value). Due to variability in local city texture between zip codes or wards within a given city, integral of the first peak may not always lead to $C_n > 1$, which is required to be able to obtain angular order parameter with a minimum configuration of 3 buildings, or 2 neighbors. If such situation exists, we proceed with the next local minimum in the distribution of $g(r)$ until $C_n > 1$ has been obtained. With such defined $C_n$ we proceed with calculations of the second city texture value, order parameter. Exact $r_{min}$ values used to define the first peak are presented in S1,S3 Tables, with $g(r)$ functions visualized in S3–S5 Figs. Applied at the city scale, we use the absolute values of $C_n$, to find $\varphi$, which characterizes the average angular distortion of buildings compared to a perfect angular local order of a city at fixed $m = C_n$ within the first shell distance determined from the integral of $g(r)$ (Eq. 7):

$$\varphi = \frac{1}{N} \sum_{j=1}^{N} \frac{1}{N_a(j)} \left| \left( \sum_{k=1}^{N_a(j)} exp\left(iC_n\vartheta_k\right) \right) \right|$$

(8)

where, $N$ is the number of buildings.

Demonstration of Eq. 2: by definition, CFR writes:

$$CFR = \frac{d}{c}$$

(9)

Where d and c are the number of reported deaths and cases, respectively, corresponding to a steady state averaged over a given period of time as explained in main part of the manuscript. Note that, in a linear response theory formulation, the number of deaths equals CFR×c. Among the recorded cases, a fraction will correspond to the at-risk population fraction as defined by the $P_i$ dimensionless parameter (see above). As a consequence:

$$c_{P_i} = (1 - P_i)\, c$$

(10)

Along the same line of reasoning, this last fraction of population can be corrected for indoors living conditions as described by the $I_i$ parameter defined above. Therefore, the fraction population not "fragile" and not in difficult indoors housing conditions but COVID-19 positive, writes:

$$c_{P_i,I_i} = (1 - I_i)(1 - P_i)\, c$$

(11)

Again, this fraction population can be corrected for the local outdoors conditions with the parameter $O_i$ which includes local city texture and weather conditions through the relative humidity. Therefore, the fraction population COVID-19 positive not "fragile", with no difficult indoors housing conditions and not living in unfavorable urban areas with respect to airborne diseases propagation, writes:

$$c_{P_i,I_i,O_i} = (1 - O_i)(1 - I_i)(1 - P_i)\, c$$

(12)

Hence

$$d_{P_i,I_i,O_i} = a_i(1 - O_i)(1 - I_i)(1 - P_i)\, c = a_0 c$$

(13)

In the above equation, $a_0$ is the intrinsic CFR affecting a population with no urban, wealth and health conditions. The model weight parameters, w's, have been optimized by minimizing the error between predicted and measured $CFR(\lambda_1)$ values using standard statistical methods. Because the FR and $a_i$ values span over several orders of magnitudes, we adopted a sum of absolute errors, $\varepsilon$, to quantify errors in regression analyses, in the form of:

$$\varepsilon = \sum_{i=1}^{n=20} \left| CFR_i\left(\lambda_1\right) - a_i\left(\lambda_1\right) \right| \tag{14}$$

where $i$ is the city district ID from S4 Table. Since CFR values are fractions, differences between any predicted and measured values lead to small fractions and thus squaring the difference would lead to even smaller values thereby introducing bias to any optimization approach that is trying to minimize the sum of errors. As such the absolute sum $\varepsilon$ provides a more stable optimization parameter. We minimized $\varepsilon$ using a non-linear generalized reduced gradient (GRD) method with constraint convergence of 10–5, forward derivates population size of 1000. With such defined parameters, we obtained optimized urban factors and intrinsic CFR, $a_0\left(\lambda_1\right)$ (Fig 3).

Developing Eq. 13 and dropping higher terms lead to:

$$d_{P_i,I_i,O_i} = a_i\left[1 - \left(O_i + I_i + P_i\right)\right]c \tag{15}$$

Interestingly, Eq. 15 does not lead to a good description of ward or zip code scale of COVID-19 CFR indicating that [Eq. 15] Eq. 2 is actually operational and predictive complementing the time-dependent SIR approach provided in ref [34].

## Significance statement

Human health in urban environments has emerged as a primary focus of sustainable development during the time of global pandemic. Potential mitigation solutions for controlling COVID-19 and other types of airborne diseases may be identified by studying the connection between the disease severity of pathogens transmitted through the air and urban environments. By utilizing urban factors, we model a probabilistic progression of COVID-19, which shows the risk and severity of the virus at multi-geospatial scales worldwide ranging from zip-codes and neighborhoods to cities and countries. This probabilistic modeling approach enables public health officials to identify high-risk areas and deploy targeted interventions, even when personal health data or widespread testing are limited. By focusing on the physical and social characteristics of cities, our work offers a practical tool for improving pandemic preparedness and response as a path towards facilitating timely deployment of targeted countermeasures and confinement strategies and highlights the importance of urban planning in protecting communities from future outbreaks.

## Supporting information

**S1 Table. Cumulative CFR data for countries for period between March 2020 and January 2022 obtained from World Health Organization (WHO).**
(TIF)

**S2 Table. Sources for data for city districts and cities.** Multiple sources for humidity were used to obtain average values presented in S5 Table.
(TIF)

**S3 Table. Sources for humidity data for city districts data used for model optimization (Fig 4a).** All sources for each location were used to derive humidity values in S4 Table.
(TIF)

**S4 Table. City Districts measured and predicted CFR data used to derive optimal Urban Factors with their input values.** District boundaries are defined using boundaries as shown in S1 Fig.
(TIF)

**S5 Table. Cities measured and predicted CFR data with Urban Factors and input values used to derive them.** *Outliers in CFR predictions – cities with CFR values close to intrinsic CFR, $a_0$. **Cities with no CFR estimates due to the lack of

publicly available COVID-19 raw data (confirmed cases and deaths). Values in () for $\rho_b$ are corrected for the discrepancies in building footprints data due to an incorrect number of buildings. n/a shows lack of publicly available data.
(TIF)

**S6 Table. Chicago IL, USA measured and predicted data with Urban Factors and input values used to derive them at the zip code level (S5a Fig), which were merged to form Chicago districts (S2d Fig).**
(TIF)

**S7 Table. London UK measured and predicted CFR data with Urban Factors and input values used to derive them at the level of boroughs (S5b Fig).**
(TIF)

**S8 Table. Measured and predicted CFR data for pneumonia and flu with sources for each location data needed to derive the values.**
(TIF)

**S1 Fig. Healthcare Access Quality factor at the nation scale for 200+ countries.** Note that the outliers are Middle East countries with a low<65yo population but with large investments in their health care system (7).
(TIF)

**S2 Fig. (a) New York NY boroughs, (b) Los Angeles CA city and county districts, (c) Chicago IL districts, (d) Seattle WA districts.** Map created using the Free and Open Source QGIS. Copyright-free countries boundaries data were taken from the world bank (https://datacatalog.worldbank.org/search/dataset/0038272).
(TIF)

**S3 Fig. CFR distribution capturedvs. the ratio of cumulative daily cases and deaths reported within the first 6–8 months of the COVID-19 pandemic during $\lambda_1$ for 5 US cities that were used to evaluate CFR for city districts (S4 Table) demonstrating the exponent in Eq. 1 is close to unity.**
(TIF)

**S4 Fig. Radial distribution function, $g(r)$, for cities in S5 Table.** Dashed vertical lines show the distance limits, $r_{min}$, used in the determination of order parameter, $\varphi$.
(TIF)

**S5 Fig. (a) Chicago IL, USA zip codes and (b) Greater London UK boroughs.** Map created using the Free and Open Source QGIS. Copyright-free countries boundaries data were taken from the world bank (https://datacatalog.worldbank.org/search/dataset/0038272).
(TIF)

**S6 Fig. Washington D.C. USA wards data showing a) geographical boundaries of wards and b) measured and predicted CFR with their input values.** Map created using the Free and Open Source QGIS. Copyright-free countries boundaries data were taken from the world bank (https://datacatalog.worldbank.org/search/dataset/0038272).
(TIF)

**S7 Fig. Urban Factors model predications showing a comparison between measured and predicted $CFR(\lambda_1)$ from Eq. 5 for 30 zip codes in Chicago IL, USA (S6 Table) and 30 boroughs in London, UK (S7 Table), 8 wards in Washington D.C. USA between March 2020 and September 2020.** Linear fitting with slope coefficient of unity provides $R^2 = 0.59$ and $RMSE = 0.027$. For comparison city and city district data are presented in this figure. Predicted CFR values used the urban factor weight parameters from Fig 3, the same as CFR values in Fig 4.
(TIF)

**S1 Text. Covid paper.**
(DOCX)

## Author contributions

**Conceptualization:** Roland J-M Pellenq, Jacob Roxon, Mircea T. Sofonea.

**Data curation:** Jacob Roxon.

**Formal analysis:** Roland J-M Pellenq, Jacob Roxon, Mircea T. Sofonea.

**Funding acquisition:** Roland J-M Pellenq, Eric Vilain.

**Investigation:** Roland J-M Pellenq, Jacob Roxon, Mircea T. Sofonea.

**Methodology:** Roland J-M Pellenq, Jacob Roxon, Mircea T. Sofonea.

**Project administration:** Roland J-M Pellenq, Marie-Sophie Dumont.

**Resources:** Roland J-M Pellenq.

**Software:** Jacob Roxon.

**Supervision:** Roland J-M Pellenq, Eric Vilain, Mircea T. Sofonea.

**Validation:** Roland J-M Pellenq, Jacob Roxon, Eric Vilain, Marie-Sophie Dumont, Mircea T. Sofonea.

**Visualization:** Roland J-M Pellenq, Jacob Roxon.

**Writing – original draft:** Roland J-M Pellenq, Jacob Roxon.

**Writing – review & editing:** Roland J-M Pellenq, Jacob Roxon, Eric Vilain, Mircea T. Sofonea.

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
