## [Decision Letter · Decision Letter 0]

PDIG-D-23-00487

How do Urban Environment and Population Characteristics Control the Severity of COVID-19? Toward Spatially-Resolved Epidemiology

PLOS Digital Health

Dear Dr. PELLENQ,

Thank you for submitting your manuscript to PLOS Digital Health. After careful consideration, we feel that it has merit but does not fully meet PLOS Digital Health's publication criteria as it currently stands. Therefore, we invite you to submit a revised version of the manuscript that addresses the points raised during the review process.

Please submit your revised manuscript within 60 days Apr 15 2024 11:59PM. If you will need more time than this to complete your revisions, please reply to this message or contact the journal office at digitalhealth@plos.org. Please include the following items when submitting your revised manuscript:

We look forward to receiving your revised manuscript.

Kind regards,

Anat Reiner-Benaim

Academic Editor

PLOS Digital Health

Journal Requirements:

1. We ask that a manuscript source file is provided at Revision. Please upload your manuscript file as a .doc, .docx, .rtf or .tex.

Additional Editor Comments (if provided):

Reviewers' comments:

Reviewer's Responses to Questions

**Comments to the Author**

1. Does this manuscript meet PLOS Digital Health’s publication criteria ? Is the manuscript technically sound, and do the data support the conclusions? The manuscript must describe methodologically and ethically rigorous research with conclusions that are appropriately drawn based on the data presented.

Reviewer #1: Yes

Reviewer #2: No

Reviewer #3: Yes

2. Has the statistical analysis been performed appropriately and rigorously?

Reviewer #1: N/A

Reviewer #2: N/A

Reviewer #3: Yes

3. Have the authors made all data underlying the findings in their manuscript fully available (please refer to the Data Availability Statement at the start of the manuscript PDF file)?

Reviewer #1: Yes

Reviewer #2: No

Reviewer #3: Yes

4. Is the manuscript presented in an intelligible fashion and written in standard English?

PLOS Digital Health does not copyedit accepted manuscripts, so the language in submitted articles must be clear, correct, and unambiguous. Any typographical or grammatical errors should be corrected at revision, so please note any specific errors here.

Reviewer #1: Yes

Reviewer #2: No

Reviewer #3: Yes

5. Review Comments to the Author

Please use the space provided to explain your answers to the questions above. You may also include additional comments for the author, including concerns about dual publication, research ethics, or publication ethics. (Please upload your review as an attachment if it exceeds 20,000 characters)

Reviewer #1: The paper has the potential for citation but needs improvements. The title should be changed to "Impact of urban environment and population characteristics on COVID-19 severity." The introduction should be improved to better explain the work's motivation and methodology characteristics. The research gap in the introduction should be expanded. Besides, compare the proposed methodology with the latest papers.

Reviewer #2: I see the manuscript is still under construction with several comments within the MS.

I recommend to enrich the literature review with other models. Some are cited below which I recommend authors to cite in their paper.

1. Daneshgar F., Chattopadhyay S. –“A Framework for Crisis Management in Developing Countries”. Intelligent Decision Technologies: an international journal (2011); 5(2): pp. 189-199. DOI: 10.3233/IDT-2011-0106.

2. Ray P., Chattopadhyay S. – “Fuzzy Awareness Model for Disaster Situations”. Intelligent Decision Technologies: an international journal [Special Issue on Intelligent Decision Making in Dynamic Environments: Methods, Architectures and Applications] (2009); 3 (1): pp. 75-82.

3. Li J., Land L.P.W, Chattopadhyay S., Ray P. ‘E-Health Readiness Framework from Electronic Health Record Perspective’. Proceedings of SIG GlobDev Annual Workshop on ICT, Paris, France October 14-17 (2008) URL: http://www.globdev.org/dev/files/9-Paper-Li-E-Health-Readiness-Revised.PDF.

I recommend the authors to resubmit the paper after addressing the internal comments and discussing other models, mentioned above.

Reviewer #3: This paper presents a probabilistic model to provide an accurate predictive tool for the COVID 19 regardless of the geographical location. The results show that although disease screening and vaccination policies to containment and lockdowns remain critical in controlling the spread of airborne diseases, urban factors should also be taken into consideration when identifying resources and planning targeted responses to mitigate the impact and severity of the virus transmitted through air.

Overall, this paper is well-written and easy to follow. The motivation and modelling methods are quite solid, despite having few concerns, noted below. Especially, I appreciate the authors’ effort on the inline comments.

# Urban factors:

The proposed probabilistic CFR models are mainly based on 3 distinctive urban factors: personal, indoor and outdoor. I am wondering how these factors are decided? It would be nice if there are some supporting evidence from existing research?

# Practical implication:

While the authors have given a thorough discussion of the model effectiveness, will it have any practical implications regarding the real-world policy making?

6. PLOS authors have the option to publish the peer review history of their article (what does this mean? ). If published, this will include your full peer review and any attached files.

**Do you want your identity to be public for this peer review?** For information about this choice, including consent withdrawal, please see our Privacy Policy .

Reviewer #1: No

Reviewer #2: Yes: Subhagata Chattopadhyay

Reviewer #3: No

---

## [Decision Letter · Decision Letter 1]

PDIG-D-23-00487R1

How do Urban Environment and Population Characteristics Control the Severity of COVID-19? Toward Spatially-Resolved Epidemiology

PLOS Digital Health

Dear Dr. PELLENQ,

Thank you for submitting your manuscript to PLOS Digital Health. After careful consideration, we feel that it has merit but does not fully meet PLOS Digital Health's publication criteria as it currently stands. Therefore, we invite you to submit a revised version of the manuscript that addresses the points raised during the review process.

Please submit your revised manuscript within 60 days Sep 14 2024 11:59PM. If you will need more time than this to complete your revisions, please reply to this message or contact the journal office at digitalhealth@plos.org. Please include the following items when submitting your revised manuscript:

We look forward to receiving your revised manuscript.

Kind regards,

Anat Reiner-Benaim

Academic Editor

PLOS Digital Health

Journal Requirements:

Additional Editor Comments (if provided):

Reviewers' comments:

Reviewer's Responses to Questions

**Comments to the Author**

1. If the authors have adequately addressed your comments raised in a previous round of review and you feel that this manuscript is now acceptable for publication, you may indicate that here to bypass the “Comments to the Author” section, enter your conflict of interest statement in the “Confidential to Editor” section, and submit your "Accept" recommendation.

Reviewer #1: All comments have been addressed

Reviewer #4: All comments have been addressed

Reviewer #5: All comments have been addressed

2. Does this manuscript meet PLOS Digital Health’s publication criteria ? Is the manuscript technically sound, and do the data support the conclusions? The manuscript must describe methodologically and ethically rigorous research with conclusions that are appropriately drawn based on the data presented.

Reviewer #1: Partly

Reviewer #4: Partly

Reviewer #5: Yes

3. Has the statistical analysis been performed appropriately and rigorously?

Reviewer #1: Yes

Reviewer #4: No

Reviewer #5: Yes

4. Have the authors made all data underlying the findings in their manuscript fully available (please refer to the Data Availability Statement at the start of the manuscript PDF file)?

Reviewer #1: Yes

Reviewer #4: Yes

Reviewer #5: Yes

5. Is the manuscript presented in an intelligible fashion and written in standard English?

PLOS Digital Health does not copyedit accepted manuscripts, so the language in submitted articles must be clear, correct, and unambiguous. Any typographical or grammatical errors should be corrected at revision, so please note any specific errors here.

Reviewer #1: Yes

Reviewer #4: Yes

Reviewer #5: Yes

6. Review Comments to the Author

Please use the space provided to explain your answers to the questions above. You may also include additional comments for the author, including concerns about dual publication, research ethics, or publication ethics. (Please upload your review as an attachment if it exceeds 20,000 characters)

Reviewer #1: I appreciate the authors' efforts in addressing previous feedback. While progress has been made, there are still areas for further enhancement in several areas. Below are detailed comments and recommendations aimed at improving the overall quality and impact of the manuscript:

Organization:

1. Structure Order:

o The paper's current structure places the Materials and Methods section at the end, which is unconventional. Typically, it should follow the Introduction to provide readers with context before detailing the methodology. Reorganize the manuscript to: Introduction, Materials and Methods, Results, Discussion, and Conclusion with Future Directions.

2. Introduction:

o The Introduction adequately frames the study's context and importance. However, it could benefit from more explicit research questions or hypotheses to guide readers through the paper's objectives.

Materials and Methods:

1. Clarity and Detail:

o While the methodology is described, there are gaps in specific details such as data collection protocols, participant selection criteria, and the tools used for analysis. Provide more clarity on these aspects to ensure reproducibility.

2. Sequence and Detailing:

o Ensure a clear and logical sequence of methods. Describe each step comprehensively, including any statistical methods or software used, to enhance transparency and reproducibility.

Results:

1. Presentation Clarity:

o Results are presented clearly, but ensure each finding is linked back to the research questions or hypotheses posed in the Introduction. This alignment strengthens the narrative flow and relevance of the findings.

2. Data Transparency:

o Consider including supplementary materials with raw data or detailed tables to support the reported results and allow for deeper scrutiny by readers and reviewers.

Discussion:

1. Interpretation and Contextualization:

o The Discussion section should not only summarize the findings but also interpret their implications in the broader context of peer assessment in virtual learning environments during the COVID-19 era. Relate findings to existing literature and theoretical frameworks to strengthen the discussion's depth.

2. Future Directions:

o Expand on the implications of the findings for future research or educational practices. Address potential limitations identified in the study and suggest avenues for further exploration to advance the field.

Conclusion:

1. Summarization:

o The Conclusion should succinctly summarize key findings and their implications without introducing new information. This section should reinforce the paper's contributions to the field.

Language and Style:

o Review the manuscript for clarity and coherence. Ensure that complex concepts are explained concisely and that language is accessible to a broad audience of educators and researchers.

References:

o Verify that all references are recent, relevant, and properly formatted according to the journal's guidelines. Ensure consistency in citation style throughout the manuscript.

Reviewer Feedback:

o Address all reviewer comments comprehensively, particularly those related to methodology clarity, theoretical grounding, and implications for practice.

Reviewer #4: How do Urban Environment and Population Characteristics Control the Severity of COVID-19? Toward Spatially-Resolved Epidemiology

Reviewer Report

Summary:

The study presents a novel approach to modeling the spatial dynamics of COVID-19 case fatality rates (CFR) using urban factors, such as personal, indoor, and outdoor activities. The authors developed a multiplicative model and optimized the model parameters using a non-linear optimization method. The model was validated on a larger dataset of 118 locations globally. Overall, the study offers a promising framework for understanding the role of urban factors in shaping COVID-19 severity, but there are several major and minor issues that need to be addressed.

Major issues:

1. Model fitting and optimization: The authors did not provide sufficient details on the convergence criteria, starting values, and sensitivity of the optimization process. The optimization was performed on a relatively small dataset, which may not be sufficient to robustly estimate the model parameters.

2. Model validation and goodness-of-fit: The authors used limited statistical metrics (R^2 and MAE) to assess the model's performance, and did not report any formal statistical tests to evaluate the significance of the model fit or the individual urban factor contributions.

3. Assumptions and limitations of the statistical approach: The authors did not address potential issues of multicollinearity or interdependence among the urban factors, nor did they consider potential spatial or temporal autocorrelation in the data.

4. Handling of outliers and missing data: The authors excluded 9 cities from the analysis without a clear justification, and did not address how they handled missing data or outliers in the dataset.

5. Generalizability and transferability of the model: The authors did not provide a clear assessment of the model's transferability to other geographical contexts or its ability to predict CFR during different stages of the pandemic.

Minor issues:

1. Interpretation and implications: The study did not provide a clear discussion of the practical implications of the model for public health decision-making and urban planning.

2. Ethical and equity considerations: The study did not address potential ethical or equity concerns related to the use of urban factors to predict disease severity and target interventions.

Other comments:

The introduction and results sections are well-written and provide a clear overview of the study's objectives and findings. The figures and tables are generally well-presented and correctly labeled.

Figures and tables: Clearly presented and correctly labeled.

Methods: Sufficiently detailed?

The methods section could be improved by providing more details on the model fitting and optimization process, as well as the handling of outliers and missing data.

Study Design: Appropriate for the question?

The study design, using a combination of publicly available data and a novel modeling approach, is generally appropriate for the research question. However, the limitations of the data and the statistical approach need to be better addressed.

Any experiments missing?

No experiments were conducted as part of this study, as it was a modeling-based analysis of existing data.

Statistics: Statistical analyses sound?

The statistical analyses have some limitations, as outlined in the major issues section. The authors should consider addressing these limitations to strengthen the reliability and robustness of the findings.

Results and data: Support conclusions?

The results generally support the conclusions, but the limitations of the model and the data need to be more thoroughly discussed.

All data accessible?

The data used in the study appears to be publicly available, but the authors should provide clear instructions or links for accessing the data.

References: Anything missing?

The reference list appears to be comprehensive and relevant to the study.

Title: Appropriate and informative?

The title is appropriate and informative, clearly conveying the focus of the study on modeling the spatial dynamics of COVID-19 case fatality rates using urban factors.

Reviewer #5: Make sure to elaborate on how the model parameters were derived and discuss their significance in more detail. Additionally, consider addressing other potential factors that may influence the Case Fatality Rate (CFR) but were not included in the model, such as healthcare infrastructure and public health policies.

7. PLOS authors have the option to publish the peer review history of their article (what does this mean? ). If published, this will include your full peer review and any attached files.

**Do you want your identity to be public for this peer review?** For information about this choice, including consent withdrawal, please see our Privacy Policy .

Reviewer #1: No

Reviewer #4: No

Reviewer #5: No

---

## [Decision Letter · Decision Letter 2]

PDIG-D-23-00487R2

How do Urban Environment and Population Characteristics Control the Severity of COVID-19? Toward Spatially-Resolved Epidemiology

PLOS Digital Health

Dear Dr. PELLENQ,

Thank you for submitting your manuscript to PLOS Digital Health. After careful consideration, we feel that it has merit but does not fully meet PLOS Digital Health's publication criteria as it currently stands. Therefore, we invite you to submit a revised version of the manuscript that addresses the points raised during the review process.

Please submit your revised manuscript within 60 days Jan 05 2025 11:59PM. If you will need more time than this to complete your revisions, please reply to this message or contact the journal office at digitalhealth@plos.org. Please include the following items when submitting your revised manuscript:

* A rebuttal letter that responds to each point raised by the editor and reviewer(s). You should upload this letter as a separate file labeled 'Response to Reviewers '. This file does not need to include responses to any formatting updates and technical items listed in the 'Journal Requirements' section below.

* A marked-up copy of your manuscript that highlights changes made to the original version. You should upload this as a separate file labeled 'Revised Manuscript with Track Changes '.

* An unmarked version of your revised paper without tracked changes. You should upload this as a separate file labeled 'Manuscript '.

We look forward to receiving your revised manuscript.

Kind regards,

Anat Reiner-Benaim

Academic Editor

PLOS Digital Health

Anat Reiner-Benaim

Academic Editor

PLOS Digital Health

Leo Anthony Celi

Editor-in-Chief

PLOS Digital Health

orcid.org/0000-0001-6712-6626

**Additional Editor Comments (if provided):**

**Reviewers' Comments:**

Reviewer's Responses to Questions

**Comments to the Author**

1. If the authors have adequately addressed your comments raised in a previous round of review and you feel that this manuscript is now acceptable for publication, you may indicate that here to bypass the “Comments to the Author” section, enter your conflict of interest statement in the “Confidential to Editor” section, and submit your "Accept" recommendation.

Reviewer #6: All comments have been addressed

2. Does this manuscript meet PLOS Digital Health’s publication criteria ? Is the manuscript technically sound, and do the data support the conclusions? The manuscript must describe methodologically and ethically rigorous research with conclusions that are appropriately drawn based on the data presented.

Reviewer #6: Partly

3. Has the statistical analysis been performed appropriately and rigorously?

Reviewer #6: Yes

4. Have the authors made all data underlying the findings in their manuscript fully available (please refer to the Data Availability Statement at the start of the manuscript PDF file)?

Reviewer #6: Yes

5. Is the manuscript presented in an intelligible fashion and written in standard English?

Reviewer #6: Yes

6. Review Comments to the Author

Reviewer #6: Review – PDIG-D-23-00483

The presented research is impressive; however, the article requires further revisions to reach a publishable standard.

The research emphasizes advanced methodology over the practical application of this approach. This focus is valid if the primary aim is to showcase the methodology, with results playing a secondary role. However, if the findings are intended to be the central focus of the article, I would recommend clarifying the direction—either concentrating solely on methodological innovation or prioritizing practical application.

The research presented is thorough and significant. However, it is challenging to read due to an overwhelming amount of continuous explanations, often requiring multiple readings of the same sentence for clarity. The frequent use of scientific notations, while familiar to the authors, in the text further complicates comprehension. Reducing the integration of notations within explanations and opting for more textual descriptions could improve readability.

Some specific comments:

1) Classifications – social sciences appear twice

2) Keywords – covid-19 and corona virus appear together, better to leave the 1st term.

3) Abstract – instead of the word "link" use connection or association.

4) The 1st paragraph of the introduction – rephrase, make sentences shorter. You have sentence spanning on 5 rows. In general, use this suggestion for the rest of the text. There are very long sentences. Even sentences that start before presenting the equations and end after the equations. The reading is not smooth.

5) Figure 1 – there is only the title of the figure that must be placed under the figure. The comments about tables s1, s2 must by in the text.

6) Was figure 1 prepared specifically for this paper? If not, please provide the reference.

7) The literature review is poor. I understand that in some journals there is no need of separate sections of the introduction and literature review, but still there are a lot of statements in the introduction section that require citations.

8) Page 4, 2nd paragraph. Is "sane individual" the right term?

9) Page 4, last paragraph. The word "provide" appears twice in the same sentence.

10) The table of all scientific notations should be presented in the materials section.

11) Page 5. What amount of data that was removed from the analysis?

12) Discussion. The definitions of 3rd and 4th waves are wrong. Each wave couldn't be 1.5-2 years.

13) In my opinion, the results part with illustrations must be much more detailed and comparative. The methods part is too detailed.

7. PLOS authors have the option to publish the peer review history of their article (what does this mean? ). If published, this will include your full peer review and any attached files.

**Do you want your identity to be public for this peer review?** For information about this choice, including consent withdrawal, please see our Privacy Policy .

Reviewer #6: None

**Figure resubmission:**
---

## [Editor Report · Decision Letter 3]

PDIG-D-23-00487R3How do Urban Environment and Population Characteristics Control the Severity of COVID-19? Toward Spatially-Resolved EpidemiologyPLOS Digital Health

 Dear Dr. PELLENQ,

 Thank you for submitting your manuscript to PLOS Digital Health. After careful consideration, we feel that it has merit but does not fully meet PLOS Digital Health's publication criteria as it currently stands. Therefore, we invite you to submit a revised version of the manuscript that addresses the points raised during the review process.

 In particular, please follow the reviewer recommendation to improve the text clarity and efficiency. English editing is advised. Please see all comments below.

 Please submit your revised manuscript within 60 days May 11 2025 11:59PM. If you will need more time than this to complete your revisions, please reply to this message or contact the journal office at digitalhealth@plos.org. 

* A rebuttal letter that responds to each point raised by the editor and reviewer(s). You should upload this letter as a separate file labeled 'Response to Reviewers '. This file does not need to include responses to any formatting updates and technical items listed in the 'Journal Requirements' section below.

* A marked-up copy of your manuscript that highlights changes made to the original version. You should upload this as a separate file labeled 'Revised Manuscript with Track Changes '.

* An unmarked version of your revised paper without tracked changes. You should upload this as a separate file labeled 'Manuscript '.

We look forward to receiving your revised manuscript.

Kind regards,

Anat Reiner-Benaim

Academic Editor

PLOS Digital Health

Anat Reiner-Benaim

Academic Editor

PLOS Digital Health

Leo Anthony Celi

Editor-in-Chief

PLOS Digital Health

orcid.org/0000-0001-6712-6626

**Journal Requirements:**

**Additional Editor Comments (if provided):**

**Reviewers' Comments:**

**Figure resubmission:**

 **Reproducibility:**  To enhance the reproducibility of your results, we recommend that authors of applicable studies deposit laboratory protocols in protocols.io, where a protocol can be assigned its own identifier (DOI) such that it can be cited independently in the future. Additionally, PLOS ONE offers an option to publish peer-reviewed clinical study protocols. Read more information on sharing protocols at https://plos.org/protocols?utm_medium=editorial-email&utm_source=authorletters&utm_campaign=protocols

---

## [Editor Report · Decision Letter 4]

Urban Environmental and Population Factors as Determinants of COVID-19 Severity: A Spatially-Resolved Probabilistic Modeling Approach

PDIG-D-23-00487R4

Dear Prof PELLENQ,

We are pleased to inform you that your manuscript 'Urban Environmental and Population Factors as Determinants of COVID-19 Severity: A Spatially-Resolved Probabilistic Modeling Approach' has been provisionally accepted for publication in PLOS Digital Health.

Best regards,

Anat Reiner-Benaim

Academic Editor

PLOS Digital Health